# Tight Bounds and Achievable Upper Bounds of Minimal Dimensions for Embedding-based Retrieval

## Abstract

This paper studies the minimal dimension required to embed subset memberships ($m$ elements and $\binom{m}{k}$ subsets of at most $k$ elements) into vector spaces, denoted as Minimal Embeddable Dimension (MED). The tight bounds of MED are derived theoretically and supported empirically for various notions of "distances" or "similarities", including $\ell_2$ metric, inner product, and cosine similarity. In addition, we conduct numerical simulation in a more achievable setting, where the $\binom{m}{k}$ subset embeddings are chosen as the centroid of embeddings of the contained elements. Our simulation easily realizes a logarithmic dependency between the MED and the number of elements to embed. These findings imply that embedding-based retrieval limitations stem primarily from learnability challenges, not geometric constraints, guiding future algorithm design.

## 1 Introduction

Embedding-based retrieval systems answer queries by vector comparison. One common setting of such systems is that the system maintains each element $x_i \in X$ as a vector $\boldsymbol{x}_i \in \mathbb{R}^d$, where $X$ is the universe set of elements and $\mathbb{R}^d$ is the vector space[1]. Each query $q$ to answer is also embedded as a vector $\boldsymbol{w}_q \in \mathbb{R}^d$. The answers to a query $q$ are retrieved by (1) comparing $\boldsymbol{w}_q$ against each of the $\boldsymbol{x}_i$ with a scoring function $s(\cdot, \cdot)$ to obtain a score $s_{i|q} = s(\boldsymbol{x}_i, \boldsymbol{w}_q)$ and (2) retrieving the answers with the $k$ largest scores $s_{i|q}$[2]. This paper investigates a fundamental question of the aforementioned retrieval systems. One informal statement is:

> Given the universe $X$ with $\underline{m\ \text{elements}}$ and a $\underline{\text{scoring function } s}$, what is the **minimal** $\underline{\text{dimension } d}$ of $\mathbb{R}^d$ such that **every** query with $\underline{\text{at most } k \text{ answers}}$ is perfectly retrievable?

This question concerns the "Minimal Embeddable Dimension (MED)" $d$, as will be formally defined in Section 2, which depends on the function $s$, the cardinality $m$ of the universe $X$, and also the cardinality $k$ of the answer set to queries we are interested in. For convenience, we say a set of embeddings for elements and subsets is an **embeddable configuration** if and only if all answers can be retrieved by score comparison.

The study of the minimal embeddable dimension is fundamentally related to several classic research directions, including the VC-dimension in statistical learning theory [Mohri et al., 2018] and the $k$-set problem in combinatorial geometry [Matousek, 2013]. It is widely acknowledged that this question is related to the fundamental boundary underlying modern information retrieval [Lee et al., 2019; Weller et al., 2025b], data compression [Andoni & Indyk, 2008], and representation learning [Izacard et al., 2021; Wang et al., 2022].

The effectiveness of embedding-based retrieval has been demonstrated to depend on the dimension $d$ of the space when the size $m$ of the universe is large [Yin & Shen, 2018; Reimers & Gurevych, 2021]. Very recently, Weller et al. [2025a] suggested that the fundamental property of the underlying geometric space limits the effectiveness of embedding-based retrieval, as supported by both theoretical

---

[1]Other metric spaces may apply. $\mathbb{R}^d$ is chosen because it is the most commonly used.

[2]Other notions of answer sets may apply, but top-$k$ is chosen also because its common usage.

Table 1: Tight bounds of minimal dimensions in the standard setting (MED) and the centroid setting (MED-C) of important scoring functions. Theorems are stated in Section 3 and Section 4. $m$ denotes the size of the universe, $k$ indicates the largest size subset to query. Interestingly, the MED does not depend on $m$. The $O(\log m)$ upper bound of MED-C can be easily observed in simulation. MED is the lower bound of MED-C by definition, so the $\Omega(k)$ lower bound of MED-C is omitted in the table.

| Scoring function | Standard setting (MED) | | Centroid setting (MED-C) |
|---|---|---|---|
| | Lower bound | Upper bound | Upper bound |
| Linear (inner product) | $k-1$ | $2k$ | $O(k^2 \log m)$ |
| Cosine similarity | $k-1$ | $2k+1$ | $O(k^2 \log m)$ |
| Euclidean distance ($\ell_2$) | $k-1$ | $2k$ | $O(k^2 \log m)$ |

findings and numerical simulations. On the theory side, they considered a retrieval problem defined by a query-relevance matrix $A \in \{0,1\}^{m \times n}$ and $s(\boldsymbol{x}_i, \boldsymbol{w}_q) = \langle \boldsymbol{x}_i, \boldsymbol{w}_q \rangle$ is the inner product[3]. They connected the minimal dimension $d$ with the sign-rank from the query-relevance matrix $A$, specifically, $\mathrm{rank}_\pm(2A - 1) - 1 \le d \le \mathrm{rank}_\pm(2A - 1)$, where the sign-rank of a matrix $M \in \mathbb{R}^{m \times n}$ is defined as $\mathrm{rank}_\pm M = \min\{\mathrm{rank}B | B \in \mathbb{R}^{m \times n}$ such that for all $i, j$ we have $\mathrm{sign}B_{ij} = M_{ij}\}$. However, they failed to explicitly construct the relationship in terms of $m$ due to the hardness of computing the sign rank. Their simulation, on the other hand, employs a *free embedding optimization* to check whether there exists an embeddable configuration of $m$ elements and $\binom{m}{k}$ subsets in $\mathbb{R}^d$. They empirically established a *polynomial* relationship between $d$ and $m$. **It is this empirical fitting result that suggested the minimal dimension $d$ required grows with the number of elements $m$ to be retrieved, and indicated the fundamental limit lies in the geometric space.**

This paper questions the conclusions by Weller et al. [2025a] by studying the MED problem, as will be formalized in Section 2, from the perspective of approximability. That means we also work on the fundamental property of geometric spaces as in Weller et al. [2025a] but with an extended set of essential scoring functions: in addition to the inner product previously discussed [Weller et al., 2025a], cosine similarity and Euclidean distances are also included due to their wide application in embedding-based systems and representation learning. This paper discusses **when those spaces are large enough** to contain the subset membership of at most $k$ elements. This perspective, due to its theoretical nature, is not practical enough to help develop scalable ML algorithms. However, *it still allows people to determine whether the effectiveness of embedding-based systems is limited by* **approximability** (a fundamental property of geometric spaces) *or by* **learnability** (the way we build specific retrieval systems). If it turns out to be an approximability issue, any methodological effort might make little progress before the dimension of the vector space is large enough. Otherwise, there is still hope in developing more performant embedding-based retrieval systems.

**Theoretical Contribution.** We establish a clear quantitative relation $d = \Theta(k)$ between the MED $d$ and the maximum cardinality $k$ of the answer set for all three scoring functions. The major results are presented in Table 1, "Standard setting (MED)". Our theoretical results are independent of the universe's size $m$, suggesting that the fundamental property of geometric space **does not** limit the embedding-based retrieval system. They sharpen the theoretical bounds by Weller et al. [2025a] and contradict the empirical results in Weller et al. [2025a].

**Empirical Support.** We also conduct numerical simulations to show that the minimal dimension, where an embeddable configuration of embeddings can be found by optimization, does not grow polynomially with $m$, but at most logarithmically. The numerical simulations also justify the correctness of our tight bounds in MED and credit the limitation of the effectiveness of embedding-based systems to learnability rather than approximability. Some brief details are presented here, and more information can be found in Section 4.

In our simulation, we **further assume** the query vector of $q$ to be given by $\boldsymbol{w}_q := \frac{1}{|S|} \sum_{x \in S} \boldsymbol{x}$ as the centroid of vectors in the answer set $S$ and denoted as the **centroid setting**. Our centroid setting is "more achievable" than the "free embedding optimization" setting [Weller et al., 2025a] because

---

[3]To present all top-$k$ subset queries, the query-relevance matrix A is in the huge space $\{0, 1\}^{\binom{m}{k} \times m}$.

our setting only requires optimizing $m$ embeddings for elements, but requires no optimization of the $\binom{m}{k}$ embeddings of the subsets. Simulations in both settings only show the existence of an embeddable configuration of $m$ elements in specific dimensions. By definition, they reveal upper bounds on MED.

Another interesting finding is that the simulation in the centroid setting reveals an empirical logarithmic upper bound of MED. This $O(\log m)$ dependency agrees with our theoretical upper bounds from a probabilistic construction. The minimal embeddable dimension in the centroid setting is denoted MED-C, and listed in Table 1. Compared to the polynomial dependency revealed by the free embedding optimization [Weller et al., 2025a], it might be counterintuitive because the centroid setting has fewer degrees of freedom ($\binom{m}{k}$ subset query embeddings are not optimized in the centroid setting), providing a perfect example of the bottleneck in learnability rather than approximability. More discussion can be found in Section 4.

In summary, our work reframes the debate from geometric limits to learnability, offering both theoretical guarantees and empirical evidence that low-dimensional embeddings suffice for perfect retrieval.

This paper is organized as follows. Section 2 formally defines the Minimal Embeddable Dimension (MED), establishes some inequalities of MED, and also introduces the centroid setting. Section 3 proves the tight bounds of the MED under various scoring function classes. Section 4 derives the bounds for MED in centroid settings and validates them through numerical simulations. Finally, Section 5 concludes the paper with limitations and future directions.

## 2 MINIMAL EMBEDDABLE DIMENSION

For convenience, we summarize the notations in this paper. $m, n, d, k$ are positive integers. $n$ and $d$ are both used for dimension. They can be used interchangeably. The subtle difference is that $d$ usually represents dimension in general while $n$ appeared in the quantitative relations. $X$ is the set of $m$ elements to be embedded and $\boldsymbol{x} \in \mathbb{R}^d$ denotes the embeddings of $x \in X$. For simplicity, we don't distinguish the embedding $\boldsymbol{x}$ and element $x$, so it is fine to state $X = \{\boldsymbol{x}_i\}_{i=1}^m \subseteq \mathbb{R}^d$. We study all possible queries $q$ that concern at most $k$ objects in $X$. In that sense, it is equivalent to consider all subsets of at most $k$ elements in $X$ ($k \leq m$ by default), denoted as $\mathcal{C}_k = \{S \subseteq X, |S| \leq k\}$ and $\mathcal{C}_m = 2^X$. We also use $q$ to denote a subset in $X$. The scoring function $s : \mathbb{R}^d \times \mathbb{R}^d \mapsto \mathbb{R}$ measures the relatedness of two vectors in $\mathbb{R}^d$. The description of $\mathbb{R}^d$ is omitted if the context is clear. The scoring functions of our interest include:

**Linear:** $s_{\text{linear}}(\boldsymbol{x}, \boldsymbol{w}) = \langle \boldsymbol{x}, \boldsymbol{w} \rangle$, where $\boldsymbol{w}$ is a query vector, $\langle \cdot, \cdot \rangle$ is the inner product.
**Cosine similarity:** $s_{\cos}(\boldsymbol{x}, \boldsymbol{w}) = \frac{\langle \boldsymbol{x}, \boldsymbol{w} \rangle}{\|\boldsymbol{x}\| \|\boldsymbol{w}\|}$.
**Euclidean distance ($\ell_2$):** $s_{\ell_2}(\boldsymbol{x}, \boldsymbol{w}) = -\|\boldsymbol{x} - \boldsymbol{w}\|_2$.

For convenience, we consider the functional classes $\mathcal{F}$ induced by those three scoring functions, e.g. The linear functional class $\mathcal{F}_{\text{linear}} = \{f(\cdot) := s_{\text{linear}}(\cdot, \boldsymbol{w}) | \boldsymbol{w} \in \mathbb{R}^d\}$, the cosine family $\mathcal{F}_{\cos} = \{f(\cdot) := s_{\cos}(\cdot, \boldsymbol{w}) | \boldsymbol{w} \in \mathbb{R}^d\}$, and the $\ell_2$ family $\mathcal{F}_{\ell_2} = \{f(\cdot) := s_{\ell_2}(\cdot, \boldsymbol{w}) | \boldsymbol{w} \in \mathbb{R}^d\}$. Each functional $f \in \mathcal{F} : \mathbb{R}^d \mapsto \mathbb{R}$. We use the suffix to indicate the specific function $f_q \in \mathcal{F}$ is used for a specific query $q$ or $f_S$ for a subset $S$.

The primary focus of this section is to present the definition and general properties of the Minimal Embeddable Dimension (MED). Notably, we dedicate the last subsection 2.3 to a special **centroid** setting, where the $\boldsymbol{w}_q = \frac{1}{|S|} \sum_{x \in S} \boldsymbol{x}$, which concerns the MED in centroid (**MED-C**), an upper bound of MED.

### 2.1 $k$-SHATTER PROBLEM

To formally define the minimal embeddable dimension, we introduce the concept of $k$-shattering.

**Definition 2.1** ($k$-shattering). Let $X \subseteq \mathbb{R}^d$ be a set of $m$ points. $X$ **is $k$-shattered by** $\mathcal{F}$ if and only if $\forall S \in \mathcal{C}_k, \exists f_S \in \mathcal{F}, \forall \boldsymbol{x} \in S, \forall \boldsymbol{y} \notin S, f_S(\boldsymbol{x}) > b_S > f_S(\boldsymbol{y})$, where $b_S \in \mathbb{R}$ depends on $S$ and $\mathcal{F}$.

*Remark* 2.2. The definition of $k$-shattering precisely determines whether there exists a configuration of vectors in $X$ under a specific functional family or scoring function with query embeddings

such that **the embedding-based retrieval built upon this configuration succeeds on all queries concerning at most $k$ elements**.

Minimal Embeddable Dimension (MED) is then defined based on $k$-shattering, which depends on $m$, $k$, and $\mathcal{F}$. For convenience, MED is denoted as a function $n^* = \text{MED}(m, k; \mathcal{F})$.

**Definition 2.3** (Minimal Embeddable Dimension). Given $m$, $k$, $\mathcal{F}$, MED $n^*$ is the integer that a configuration of $m$ points that can be $k$-shattered by $\mathcal{F}$ exists in $\mathbb{R}^{n^*}$ but not in $\mathbb{R}^{(n^*-1)}$.

One direct result, according to Definition 2.3, is the non-strict monotonicity of $\text{MED}(m, k; \mathcal{F})$.

**Proposition 2.4.** *For $2 \leq k \leq m$, the following inequality holds:*

$$\text{MED}(m, k-1; \mathcal{F}) \leq \text{MED}(m, k; \mathcal{F}) \leq \text{MED}(m+1, k; \mathcal{F}). \tag{1}$$

Meanwhile, when all subsets of at most half the points can be shattered, all subsets can be shattered.

**Proposition 2.5.** *For $k \geq \lfloor \frac{m}{2} \rfloor$, $\text{MED}(m, k; \mathcal{F}) = \text{MED}(m, m; \mathcal{F})$.*

## 2.2 General bounds of MED by VC dimension

The definition of $k$-shattering also defines the VC dimension [Mohri et al., 2018], which is redefined below in terms of $k$-shattering.

**Definition 2.6** (VC dimension). The VC dimension of a functional family $\mathcal{F}$ (maps $\mathbb{R}^n$ to $\mathbb{R}$) is the maximal size $m$ of a set $X$ that can be $m$-shattered by $\mathcal{F}$, denoted as $m = \text{VCD}(n; \mathcal{F})$.

VC dimension is a measure of the capacity of classes of sets or binary functions [Vapnik, 2013]. This concept plays a fundamental role in statistical learning theory, particularly in the study of approximability. By defining both the MED and VC dimensions using $k$-shattering, it is reasonable to expect some connections to hold between them, and indeed they do. For notational convenience, we define the inversion of VC dimension $m = \text{VCD}(n; \mathcal{F})$ as $n = \text{VCD}^{-1}(m; \mathcal{F})$.

**Lemma 2.7.** *If $m = \text{VCD}(n; \mathcal{F})$, then $\text{VCD}^{-1}(m-1) < \text{MED}(m, m; \mathcal{F}) \leq \text{VCD}^{-1}(m)$.*

*Proof.* Given $m = \text{VCD}(n, \mathcal{F})$, then (1) $m$ points in $\mathbb{R}^n$ can be $m$-shattered by $\mathcal{F}$ but (2) $m + 1$ points in $\mathbb{R}^n$ cannot be $(m+1)$-shattered by $\mathcal{F}$. Those two claims imply, in the sense of MED, that (1) $\text{MED}(m, m, \mathcal{F}) \leq n$ and (2) $\text{MED}(m+1, m+1, \mathcal{F}) > n$. The desired inequalities can be proved by substitutions. $\square$

Additionally, combining Proposition 2.4 and Lemma 2.7 yields the following proposition.

**Proposition 2.8.** $\text{VCD}^{-1}(k-1; \mathcal{F}) < \text{MED}(m, k; \mathcal{F}) \leq \text{VCD}^{-1}(m; \mathcal{F})$.

Thus, a rough lower and upper bound of MED can be derived from the VC dimension. Specifically, the upper (lower) bounds of VC dimensions now form the lower (upper) bounds of MED, respectively.

## 2.3 MED in the centroid setting

We present the centroid setting, where the query embedding is the simple average of the element embeddings. The centroid setting imposes more constraints than $k$-shattering. Formal definitions are now provided.

We begin with the definition of $k$-centroid shattering as the restricted version of $k$-shattering.

**Definition 2.9** ($k$-centroid shattering). Let $X \subseteq \mathbb{R}^d$ be a set of $m$ points, $X$ is $k$-centroid shattered by a scoring function $s(\cdot, \cdot)$ if and only if $\forall S \in \mathcal{C}_k, \forall \boldsymbol{x} \in S, \forall \boldsymbol{y} \notin S, s(\boldsymbol{x}, \boldsymbol{c}_S) > s(\boldsymbol{y}, \boldsymbol{c}_S)$, where $\boldsymbol{c}_S = \sum_{x \in S} \frac{1}{|S|} \boldsymbol{x}$ is the center vector of $S$.

*Remark* 2.10. We stress that the concept of centroid shattering concerns a specific scoring function ($s_{\text{linear}}, s_{\text{cos}}, s_{\ell_p}$) rather than a functional class $\mathcal{F}_{\text{linear}}, \mathcal{F}_{\text{cos}}, \mathcal{F}_{\ell_p}$ because we already fix the vector for each of the query as the center of its answer set $S$, specifically, $\boldsymbol{w} := \boldsymbol{c}_S$.

Then, the MED in the centroid setting is defined below,

**Definition 2.11** (MED in Centroids (MED-C)). Given $m, k, s(\cdot, \cdot)$, MED-C $n^* = \text{MED-C}(m, k; s)$ is the integer that a configuration of $m$ points that can be $k$-centroid shattered by $s(\cdot, \cdot)$ exists in $\mathbb{R}^{n^*}$ but not in $\mathbb{R}^{(n^*-1)}$.

**MED-C and MED.** In the original definition of MED, only the existence of functionals is required to prove the $k$-shattering definition. In the centroid setting, $k$-centroid shattering fixes the subset query embeddings $\boldsymbol{c}_S$, which further determine functionals explicitly. Noticing that the number of "free" functionals involved in $k$-shattering is $\binom{m}{k}$, the $k$-centroid setting reduces the freedom into as few as $m$ vectors, which facilitates numerical simulation. For the same reason, however, MED lower bounds MED-C because of the flexibility to freely determine $f(\cdot)$ other than the centroid one $s(\cdot, \boldsymbol{c}_S)$. Formally, we have the following proposition.

**Proposition 2.12.** $\text{MED}(m, k; \mathcal{F}) \leq \text{MED-C}(m, k; s)$, where $\mathcal{F}$ is the functional family induced by the scoring function $s$.

*Proof.* Let $n = \text{MED-C}(m, k; s)$, we consider the configuration of vectors $\boldsymbol{x}_i$ for $i = 1, ..., m$ elements. Let the functionals for each $S \in \mathcal{C}_k$, we select the functional $f_S(\cdot) := s(\cdot, \boldsymbol{c}_S)$, where $\boldsymbol{c}_S = \frac{1}{|S|} \sum_{x \in S} \boldsymbol{x}$. We can say that this configuration is also $k$-shattered by $\mathcal{F}$. Then, $\text{MED}(m, k; \mathcal{F}) \leq \text{MED-C}(m, k; s)$ by definition. $\qquad \square$

**MED-C and neural set embeddings.** One of the common deep learning approaches to compute a set embedding is to aggregate the embeddings of contained elements [Zaheer et al., 2017]. We can also study the Minimum Embeddable Dimension in the Neural set embedding setting (MED-N), where the set embedding $\boldsymbol{w}_S = \text{NN}(\{\boldsymbol{x} : x \in S\})$ is computed by a complex neural architecture (such as MLP or multi-head self attention). The centroid setting simplifies the aggregation of complex neural networks by simple averaging. Therefore, one could expect that the MED-C is not smaller than the MED-N due to a less capable averaging operation. Combined with the definition of MED, this informal discussion suggested that we can roughly consider MED-N to be between MED and MED-C.

## 3 TIGHT BOUNDS FOR MED AND DISCUSSIONS

This section derives tight bounds of MED under $\mathcal{F}_{\text{linear}}$, $\mathcal{F}_{\cos}$, and $\mathcal{F}_{\ell_2}$. The choices are based on the broad interest of their applications in modern machine learning and representation learning. By Proposition 2.5, we only need to study the cases where $2 \leq k \leq \lfloor \frac{m}{2} \rfloor$. The general proof strategy applied here is to find the upper bound of MED by construction, and to find the lower bound by VC dimension by Proposition 2.8.

### 3.1 TIGHT BOUNDS OF $\text{MED}(m, k; \mathcal{F}_{\text{linear}})$

To begin with, let's consider the cyclic polytope [Ziegler, 2012].

**Example 3.1** (Cyclic polytope). Given moment curve $\boldsymbol{x}(t) = (1, t, t^2, ..., t^d) \in \mathbb{R}^d, 0 \leq t \leq 1$, a cyclic polytope is the convex hull $\text{Conv}(\boldsymbol{x}(t_1), \boldsymbol{x}(t_2), \ldots, \boldsymbol{x}(t_m))$.

One of the most well-known results of cyclic polytopes is that a cyclic polytope in $\mathbb{R}^d$ is an $\lfloor \frac{d}{2} \rfloor$-neighborly polytope, in which every $k$ vertices form a face when $k \leq \lfloor \frac{d}{2} \rfloor$ [Ziegler, 2012]. By forming a $k$-face, it means that the $k$ vertices in this face can be linearly separated from the rest of the vertices of the polytope.

**Theorem 3.2.** $k - 1 \leq \text{MED}(m, k; \mathcal{F}_{\text{linear}}) \leq 2k$.

*Proof.* Noticing the VC dimension of $\mathcal{F}_{\text{linear}}$ in $\mathbb{R}^n$ is $n + 1$ [Mohri et al., 2018], the lower bound is derived as a direct result of Proposition 2.8. The upper bound is by the cyclic polytope construct in Example 3.1. $\qquad \square$

We see that $\text{MED}(m, k, \mathcal{F}_{\text{linear}})$ only depends on $k$. Ignoring the coefficient, $\text{MED}(m, k; \mathcal{F}_{\text{linear}}) = \Theta(k)$ also holds. Then, we show that the $\mathcal{F}_{\cos}$ and $\mathcal{F}_{\ell_2}$ share the similar bounds as $\mathcal{F}_{\text{linear}}$.

## 3.2 Tight bounds of $\text{MED}(m, k; \mathcal{F}_{\ell_2})$

By geometric constructions in $\mathbb{R}^n$ regarding the $k$-shattering, the following relation is revealed.

**Proposition 3.3.** $\text{MED}(m, k; \mathcal{F}_{\ell_2}) \leq \text{MED}(m, k; \mathcal{F}_{\text{linear}})$.

*Proof.* Given a configuration that can be $k$-shattered by $\mathcal{F}_{\text{linear}}$, then for each set $S, |S| \leq k$, there exists a hyperplane $H_S$ that separates $S$ from $X - S$. We can always find an $\ell_2$ ball that includes $S$ and is tangent to $H_S$, which automatically does not contain $X - S$. □

When considering the Proposition 3.3, we conclude $\text{MED}(m, k; \mathcal{F}_{\ell_2})$ in Theorem 3.4 with additional information that $\text{VCD}(n; \mathcal{F}_{\ell_2}) = n + 1$ and Proposition 2.8.

**Theorem 3.4.** $k - 1 \leq \text{MED}(m, k; \mathcal{F}_{\ell_2}) \leq 2k$.

## 3.3 Tight bounds of $\text{MED}(m, k; \mathcal{F}_{\cos})$

**Proposition 3.5.** $\text{MED}(m, k; \mathcal{F}_{\text{linear}})) \leq \text{MED}(m, k; \mathcal{F}_{\cos}) \leq \text{MED}(m, k; \mathcal{F}_{\text{linear}})) + 1$.

*Proof sketch:* Notice that the decision boundary by cosine similarity on a sphere is the intersection between hyperplanes and this sphere. Given a configuration in $\mathbb{R}^n$ that can be $k$-shattered by $\mathcal{F}_{\text{linear}}$, one could use the inverse stereographic projection to project onto a sphere in $\mathbb{R}^{n+1}$, where the original points are projected to the points on the sphere. Meanwhile, suppose a configuration can be $k$-shattered by cosine similarities. In that case, one can first project every point onto the sphere by $x \mapsto \frac{x}{\|x\|_2}$; then, the decision boundaries on spheres can be extended to hyperplanes. The full proof can be found in Appendix A.1.

Combination of the Theorem 3.2 and the Proposition 3.5 also shows that $\text{MED}(m, k; \mathcal{F}_{\cos}) = \Theta(k)$.

**Theorem 3.6.** $k - 1 \leq \text{MED}(m, k; \mathcal{F}_{\cos}) \leq 2k + 1$.

## 3.4 Discussion on $\Theta(k)$ bound of MEDs

To summarize, the minimum embeddable dimension for $\mathcal{F}_{\text{linear}}$, $\mathcal{F}_{\cos}$, and $\mathcal{F}_{\ell_2}$ are all $\Theta(k)$, which is independent of the number of total elements in the set system. This result suggests that, despite the difficulty, it is possible to encode all top-$k$ queries into a space of no more than $2k$ dimensions, regardless of the number of elements to embed.

**The real performance bottleneck of the embedding-based retrieval**. The bounds clearly show that the geometric space **does not** limit the effectiveness of the embedding-based retrieval system. To make this even more transparent, one possible strategy to build a perfect top-$k$ queries of a retrieval system with $m$ elements is to (1) put elements in the vertices of a cyclic polytope in $\mathbb{R}^{2k}$, (2) learn the query embeddings model which predict the query embedding that clearly separates the (at most $k$) answers from the rest. Our theory guarantees that such query embeddings exist, but the real problem is how to build a function to predict them. Knowing that neural networks can universally approximate functions [Hornik et al., 1989], the bottleneck of embedding-based retrieval is essentially a learnability problem: how to actually learn such a function with embedding dimension $2k$ from data.

**Can the $\Theta(k)$ bounds be observed?** Although the fundamental property of the underlying geometric space does not put any restrictions on embedding-based retrieval. However, two issues still challenge whether we can observe this bound in the real world. (1) **limited numerical precision:** Even with the cyclic polytope example, it is still questionable when $m$ embeddings are represented in a fixed-digit floating-point number. The budget of float points does not have infinite capacity. (2) **infeasible number of functionals to determine:** To realize $\Theta(k)$ bound, one would search for $\binom{m}{k}$ functionals. The total number of tasks is infeasible to complete. Those two issues

## 4 Upper bounds of MED-C, Simulation, and Implications

This section discusses the minimal embeddable dimension in the centroid setting (MED-C). We will see that MED-C grows logarithmically with the universe's size $m$ in both theory and experiments.

### 4.1 UPPER BOUNDS

For inner product, cosine similarity, and Euclidean distance, we use the probabilistic method as a common proving strategy to prove the logarithmic upper bound. In short, we sample $m$ element embeddings from a probabilistic distribution in $\mathbb{R}^n$ and find the conditions on $k$, $m$, and $n$ such that the desired $k$-centroid shattering configuration occurs with positive probability. The random vectors here are part of the probabilistic construction and do not directly link to the simulation.

**Theorem 4.1.** MED-C$(m, k; s_{\text{linear}}) = O(k^2 \log m)$

*Proof sketch:* We consider the probabilistic method, where $v_1, \ldots, v_m \sim \mathcal{N}(0, 1/n)$ in $\mathbb{R}^n$, we show there is positive probability such that for any subset $S$, $|S| \leq k$, $v_1 \in S$ and $v_2 \notin S$ $\langle v_1, \sum_{u \in S} u \rangle > \langle v_2, \sum_{u \in S} u \rangle$ under the condition of $n > Ck^2 \log m$. The full proof can be found in Appendix A.2.

Following a similar probabilistic construction, we can also prove that $\mathcal{F}_{\cos}$ and $\mathcal{F}_{\ell_2}$ share the same upper bound. The proofs are also in the Appendix A.2.

**Theorem 4.2.** MED-C$(m, k; \mathcal{F}_{\cos}) = O(k^2 \log m)$.

**Theorem 4.3.** MED-C$(m, k; \mathcal{F}_{\ell_2}) = O(k^2 \log m)$.

Although the $O(k^2 \log m)$ bound may not be tight, it still suffices to make significant implications. An easy lower bound for MED-C is the lower bound of MED, which is $\Omega(k)$. We can see that the gap between the lower and upper bounds is a factor of $O(\log m) \times O(k)$. We discuss those two factors separately. For the factor of $O(\log m)$, it still demonstrates that a sublinear growth for MED-C with the size $m$ of the universe is feasible. For more complex neural set embedding settings, such as MED-N discussed in Section 2.3, a similar upper bound also applies. For the additional factor $O(k)$, we would like to convince the readers that $O(k \log m)$ is not very easy to achieve, so the $O(k^2 \log m)$ is not very bad. To the best of the authors' knowledge, the most similar (but not exact) problem setting to MED-C is 1-bit compress sensing [Aksoylar & Saligrama, 2014], with a lower bound of $\Omega(k \log \frac{m}{k})$, which cancels the additional $O(k)$ factor (compare $O(k^2 \log m)$ for MED-C with $\Theta(k)$ for MED) but requires an additional sparse recovery algorithm rather than simply comparing scores independently. This related work suggests that the extra cost should be paid to achieve the bound $O(k \log m)$ of MED-C.

### 4.2 NUMERICAL SIMULATION

The situation where $k \ll m$ is of particular interest because there are many documents to be retrieved, and only a very small subset is required in a real-world scenario. Therefore, numerical simulation typically focuses on large $m$ and small $k$ to demonstrate the effectiveness of the bounds. In the numerical simulation, we use "critical" to denote the largest possible number of objects for a given dimension, or the smallest dimension required to accommodate a specified number of objects.

**Experiment settings.** We use optimize $m$ embeddings randomly initialized in by standard normal distribution in $\mathbb{R}^d$. The objective function is a hinge loss computed over all pairwise positive and negative pairs across the top-$k$ queries. We use the Adam optimizer [Kingma, 2014] with a learning rate of 1. The optimization stops either after 1,000 steps or after every query finds its perfect answers. To facilitate full reproducibility, we list our Python code in the Appendix to disclose other details.

**Comparison with baselines** Previous work [Weller et al., 2025a] didn't provide a quantitative relation in their theory, but instead offered an empirical relation via optimization of free embeddings, which corresponds to the standard setting in this paper. They search for the critical cardinality of the universe $m^*$ in a fixed dimension $d$ for the $k = 2$ case, and established the following empirical results. We list their curve in Equation 2.

$$m_{\text{WBNL}}(d) = -10.5322 + 4.0309d + 0.0520d^2 + 0.0037d^3. \tag{2}$$

This result yields a pessimistic implication: the largest number of objects we can support grows with $d$ only cubically. However, our bounds in MED-C suggests that the number of objects grows exponentially with $d$.

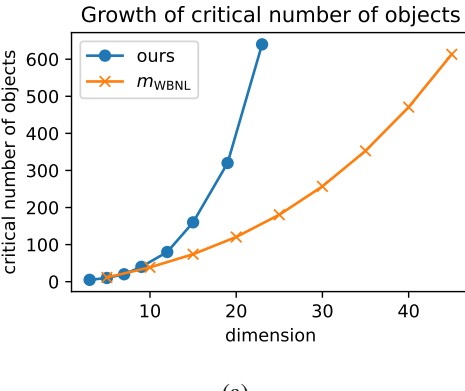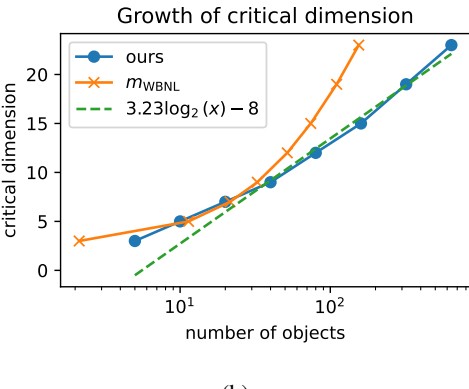

(a)  (b)

Figure 1: (a) The comparison of the growth of the critical number of points in our simulation and the curve fitted in Equation 2.; (b) The comparison of the growth of the critical dimensions in our simulation and the curve fitted in Equation 2, and the $x$ axis is plotted in a log scale.

It is a crucial question: how can the critical points identified by simulation in the centroid setting be compared with those found in the free embedding optimization setting [Weller et al., 2025a]. The goal of **numerical simulation** is to estimate the minimum dimension such that $m$ elements can be $k$-shattered in some sense (or the maximum number of elements can be located in a given dimension). Therefore, all critical points found by any numerical simulations only suggest an upper bound of the minimal dimension (or lower bound of the maximum number of elements). The goal of **this paper** is to show that the MED (or the lower bound) grows slowly enough so that the fundamental property of the geometric space does not bottleneck the embedding-based retrieval. Therefore, results from simulations, whether computed from a centroid setting or a free embedding optimization setting, can serve as empirical evidence of the MED upper bound.

To compare with their empirical results, we also ran the experiments with $k = 2$ on the centroid setting. We plot the results in Figure 1a and Figure 1b. Figure 1a suggests that the number of critical points found in our centroid setting surpasses the curve fitted by Weller et al. [2025a] easily. From Fig 1b, we can see from the results that our theory of MED-C agrees well with the empirical log-linear fitting, which is of a lower complexity family than the fitted result in Weller et al. [2025a].

### 4.3 PRACTICAL IMPLICATION

**Regarding the bottleneck of the embedding-based retrieval.** The empirical results above suggested that the actual limitation of embedding-based retrieval, i.e., the number of objects that can be embedded, actually grows at least exponentially with the dimension. This contradicts the previous claims made by Weller et al. [2025a]: The stagnation of the performance in the embedding-based system does NOT necessarily come from the fundamental geometric property of the vector space. The empirical findings resonated with the conclusion from $\Theta(k)$ bounds in a weaker form: we can observe the exponential growth of critical points and logarithmic growth of MED, but not a vertical line or horizontal line.

**Why numerical simulation in centroid setting achieves stronger points than free embedding optimization?** It is also a little bit counterintuitive that, optimizing with $\Theta(\binom{m}{k})$, more degrees of freedom actually leads to weaker results. Specifically, one should expect the optimums of the free embedding optimization and centroid setting to follow the following inequality:

$$\overbrace{\min_{\boldsymbol{w}_S:S\in\mathcal{C}_k}}^{\text{additional variables}} \min_{\boldsymbol{x}:x\in X} \sum_{S\in\mathcal{C}_k, x\in S, y\notin S} \max(0, s(\boldsymbol{y}, \boldsymbol{w}_S) - s(\boldsymbol{x}, \boldsymbol{w}_S)) \quad \text{(free embedding setting)} \quad (3)$$

$$\leq \min_{\boldsymbol{x}:x\in X} \sum_{S\in\mathcal{C}_k, x\in S, y\notin S} \max(0, s(\boldsymbol{y}, \frac{1}{|S|}\sum_{x\in S}\boldsymbol{x}) - s(\boldsymbol{x}, \frac{1}{|S|}\sum_{x\in S}\boldsymbol{x})) \quad \text{(centroid setting).} \quad (4)$$

However, the simulation empirically reveals the inequality in the other direction. It is another perfect example of how optimization affects the final results and further distorts the empirical findings.

## 5 CONCLUSION AND FUTURE WORK

This paper establishes the theoretical bounds for the minimal embeddable dimensions and studies an additional centroid setting. It revealed a positive fact: we can use a small number of dimensions to support an extensive retrieval system, provided we consider only queries with small cardinality. Meanwhile, we can clearly conclude that the effectiveness of embedding-based retrieval systems is limited by **learnability** (the way we build the specific retrieval system) and debunk the claim from the **approximability** that the theoretical limitation lies in the fundamental property of the vector space that prevents the subset membership structure from being approximated.

Future work should include more discussion of advanced embedding spaces, such as hyperbolic and Wasserstein spaces; a more sophisticated structure for queries and their answers, including situations where the cardinality of answer sets follows a power-law distribution; and a more realistic setting of fixed-point float numbers, such as float point 32, or even float point 8.

## 6 ETHICS STATEMENT

This paper studies the minimal dimensions for embedding-based retrieval, which holds the potential to help the development of the retrieval system. This is a highly theory-focused task that mainly contains mathematical proposition and their proof. There's no harm to humans done during the crafting of this paper. However, this work may lead to unexpected negative societal impact which we are unable to foresee in the current stages.

## 7 REPRODUCIBILITY STATEMENT

Regarding the experiment in this paper, the data are purely synthetic, and the code has been explicitly shown in the Appendix. Therefore, we believe it's sufficient for readers to conveniently reproduce the experiment in this paper. Regarding the theoretical part, their proofs have been explicitly given, or the related literature has been clearly cited. We believe readers can check our theoretical results with smooth.

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

# Supplementary Material

## A  ADDITIONAL PROOFS

### A.1  PROOF OF PROPOSITION 3.5

*Proof.* Let $n = \text{MED}(m, k; \mathcal{F}_{\cos})$ and there exists $X = \{x_1, \ldots, x_m\} \subset \mathbb{R}^n \setminus \{0\}$ [4]that is $k$-shattered by $\mathcal{F}_{\cos}$. Define the radial projection $\rho : \mathbb{R}^n \setminus \{0\} \to S^{n-1}$ by $\rho(x) = \frac{x}{\|x\|_2}$, and let $Y = \{\rho(x_i)\}_{i=1}^m \subset S^{n-1}$.

By $k$-shattering, for each $S \subseteq X$ with $|S| \leq k$ there exist $\boldsymbol{w}_S \in \mathbb{R}^n \setminus \{0\}$ such that

$$x \in S \Rightarrow r_S - \frac{\langle \boldsymbol{w}_S, x \rangle}{\|\boldsymbol{w}_S\|\|x\|} \leq 0, \qquad x \in X \setminus S \Rightarrow r_S - \frac{\langle \boldsymbol{w}_S, x \rangle}{\|\boldsymbol{w}_S\|\|x\|} > 0.$$

For $z \in S^{n-1}$ define the affine functional

$$f_S(z) := \left\langle \frac{-\boldsymbol{w}_S}{\|\boldsymbol{w}_S\|}, z \right\rangle + r_S \in \mathcal{F}_{\text{linear}}.$$

Then, for every $x \in X$ we have

$$f_S(\rho(x)) = r_S - \frac{\langle \boldsymbol{w}_S, x \rangle}{\|\boldsymbol{w}_S\|\|x\|},$$

so the inequalities above becomes

$$y \in \rho(S) \Rightarrow f_S(y) \leq 0, \qquad y \in Y \setminus \rho(S) \Rightarrow f_S(y) > 0.$$

Therefore we show that

$$\text{MED}(m, k; \mathcal{F}_{\text{linear}}) <= \text{MED}(m, k; \mathcal{F}_{\cos})$$

Conversely, let $n^\star = \text{MED}(m, k; \mathcal{F}_{\text{linear}})$. Then there exists $X = \{x_1, \ldots, x_m\} \subset \mathbb{R}^{n^\star}$ such that for every $S \subseteq X$ with $|S| \leq k$ there is $f_S(x) = \langle \boldsymbol{w}_S, x \rangle + b_S$ satisfying

$$x \in S \Rightarrow f_S(x) \leq 0, \qquad x \in X \setminus S \Rightarrow f_S(x) > 0.$$

Consider the embedding $\phi : \mathbb{R}^{n^\star} \to S^{n^\star} \subset \mathbb{R}^{n^\star+1}$ given by

$$\phi(x) = \frac{(x, 1)}{\|(x, 1)\|},$$

where $(x, 1)$ denotes the concatenation of $x$ and 1.

Let $Y = \{\phi(x_i)\}_{i=1}^m \subset S^{n^\star}$. For each $S$, set

$$u_S = \frac{(\boldsymbol{w}_S, b_S)}{\|(\boldsymbol{w}_S, b_S)\|} \in S^{n^\star}, \qquad t_S := 0,$$

and define $g_S(y) = \frac{\langle u_S, y \rangle}{\|u_S\|\|y\|} - t_S = \langle u_S, y \rangle$ (since $\|u_S\| = \|y\| = 1$ on $S^{n^\star}$). Then for every $x \in \mathbb{R}^{n^\star}$,

$$g_S(\phi(x)) = \left\langle \frac{(\boldsymbol{w}_S, b_S)}{\|(\boldsymbol{w}_S, b_S)\|}, \frac{(x, 1)}{\|(x, 1)\|} \right\rangle = \frac{\langle \boldsymbol{w}_S, x \rangle + b_S}{\|(\boldsymbol{w}_S, b_S)\| \cdot \|(x, 1)\|}.$$

Therefore, for $x \in X$ we have the exact shattering

$$f_S(x) \leq 0 \iff g_S(\phi(x)) \leq 0, \qquad f_S(x) > 0 \iff g_S(\phi(x)) > 0.$$

$\square$

---

[4]If there is $x_i = 0$, the proof is basically the same, we omit this situation for convenience.

## A.2 PROOF OF THEOREM 4.1, THEOREM 4.2, AND THEOREM 4.3

Those three proofs are very similar, so they are grouped together.

*Proof.* Let $v_1, \ldots, v_m \sim \mathcal{N}(0, I_n/n)$ in $\mathbb{R}^n$, our goals are to show that there is a positive probability such that for any subset $S$, $|S| \leq k$, $v_1 \in S$ and $v_2 \notin S$, the following inequalities holds:

$$\langle v_1, \sum_{u \in S} u \rangle > \langle v_2, \sum_{u \in S} u \rangle, \qquad \text{for Theorem 4.1;} \qquad (5)$$

$$\langle \frac{v_1}{\|v_1\|}, \sum_{u \in S} u \rangle > \langle \frac{v_2}{\|v_2\|}, \sum_{u \in S} u \rangle, \qquad \text{for Theorem 4.2;} \qquad (6)$$

$$\|v_1 - \frac{1}{|S|} \sum_{u \in S} u\|_2^2 < \|v_2 - \frac{1}{|S|} \sum_{u \in S} u\|_2^2, \qquad \text{for Theorem 4.3.} \qquad (7)$$

To show this, considering the concentration of the inner product of two independent random vectors $v_i$ and $v_j$ drawn from $\mathcal{N}(0, \frac{1}{n})$:

$$\Pr\left( |\langle v_i, v_j \rangle| \geq \frac{1}{3k} \right) \leq 2 \exp\left( -c\frac{n}{k^2} \right), \qquad (8)$$

where $c$ is a constant.

Also, noticing that the norm of such vectors is concentrated as

$$\Pr\left( |\|v_i\| - 1| \geq \frac{1}{3k} \right) \leq 2 \exp\left( -c\frac{n}{k^2} \right), \qquad (9)$$

**To prove theorem 4.1**, let $k = l$. For $m$ vectors, take the union bound of (1) inner products of all possible pairs and (2) the norm of all vectors, then we derive the probability that any of the inner products (or the $\|v_i - 1\|$) is greater than $\frac{1}{3k}$, and then force it to be smaller than 1.

$$2\left( m + \frac{m(m-1)}{2} \right) \exp\left( -c\frac{n}{k^2} \right) < 1. \qquad (10)$$

The probability of the event where any inner products are smaller than $\frac{1}{3k}$, and norms are no diverge from 1 larger than $\frac{1}{3k}$ is positive when

$$n > Ck^2 \log m. \qquad (11)$$

where $C$ is a constant.

Under such conditions,

$$LHS = \langle v_1, \sum_{u \in S} u \rangle = \|v_1\| + \sum_{u \in S-\{v_1\}} \langle v_1, u \rangle > 1 - \frac{1}{3k} + (|S| - 1)\frac{1}{3k} = 1 - \frac{|S|}{3k}, \qquad (12)$$

$$RHS = \sum_{u \in S} \langle v_2, u \rangle > \frac{|S|}{3k}. \qquad (13)$$

Because $|S| \leq k$, so $LHS > RHS$, Theorem 4.1 is proved.

**To prove Theorem 4.2**

$$LHS = \langle \frac{v_1}{\|v_1\|}, \sum_{u \in S} u \rangle > \frac{1 - |S|/3k}{1 + 1/3k} = \frac{3k - |S|}{3k + 1} \qquad (14)$$

$$RHS = \langle \frac{v_2}{\|v_2\|}, \sum_{u \in S} u \rangle < \frac{|S|}{3k(1 - \frac{1}{3k})} = \frac{|S|}{3k - 1}. \qquad (15)$$

Then,

$$LHS - RHS > \frac{3k(3k - 2|S| - 1)}{9k^2 - 1}. \qquad (16)$$

Because $1 < |S| \leq k$, so $LHS - RHS > 0$ and Theorem 4.2 is proved.

**To prove Theorem 4.3**

$$LHS = \|v_1 - \frac{1}{|S|}\sum_{u \in S} u\|_2^2 = \|v_1\|_2^2 + \|\frac{1}{|S|}\sum_{u \in S} u\|_2^2 - \frac{2}{|S|}\langle v_1, \sum_{u \in S} u \rangle \tag{17}$$

$$< 1 + \frac{1}{3k} - \frac{2}{|S|}(1 - \frac{|S|}{3k}) + \|\frac{1}{|S|}\sum_{u \in S} u\|_2^2, \tag{18}$$

$$RHS = \|v_2 - \frac{1}{|S|}\sum_{u \in S} u\|_2^2 = \|v_2\|_2^2 + \|\frac{1}{|S|}\sum_{u \in S} u\|_2^2 - \frac{2}{|S|}\langle v_2, \sum_{u \in S} u \rangle \tag{19}$$

$$> 1 - \frac{1}{3k} - \frac{2}{|S|}\frac{|S|}{3k} + \|\frac{1}{|S|}\sum_{u \in S} u\|_2^2. \tag{20}$$

Then,

$$LHS - RHS < \frac{2}{3k} + \frac{2}{3k} - \frac{2}{|S|}(1 - \frac{|S|}{3k}) = \frac{2}{k} - \frac{2}{|S|}. \tag{21}$$

Because $|S| \leq k$, so $LHS - RHS < 0$ and Theorem 4.3 is proved. $\square$

## B  SIMULATION CODE

```python
import torch
from torch import optim
from tqdm import trange
import itertools
import matplotlib.pyplot as plt
import numpy as np

class Trainer:
    def __init__(self, n, k):
        self.n = n
        self.k = k
        self.device="cuda" if torch.cuda.is_available() else "cpu"
        print("Using device", self.device)
        self.all_combinations = list(itertools.combinations(range(self.n)
            , self.k))
        self.subset_indices_tensor = torch.tensor([list(s) for s in self.
            all_combinations], device=self.device)
        self.subset_excluded_tensor = torch.tensor(
            [
                [i for i in range(self.n) if i not in subset_indices]
                for subset_indices in self.all_combinations
            ], device=self.device
        )
        self.total_violations = len(self.all_combinations) * self.k * (
            self.n-self.k)

    def calculate_loss(self):
        # Convert batch_subset_indices to a tensor for efficient indexing
        # Calculate sums of vectors for each subset in the batch
        # Shape: (batch_size, d)
        subset_sums = self.vector_embeddings[self.subset_indices_tensor].
            mean(dim=1)
        # Calculate dot products of subset sums with all vectors
        # Shape: (batch_size, n)
        dot_products_all = torch.matmul(subset_sums, self.
            vector_embeddings.T)
        # Get dot products with vectors in the subset by indexing
            dot_products_all
```

```python
            # Shape: (batch_size, k)
            dot_products_xi = torch.gather(dot_products_all, 1, self.
                subset_indices_tensor)
            # For each subset in the batch, calculate dot products with
                vectors NOT in the subset
            # We can create a mask to zero out elements within the subset
            # This is faster than iterating through the batch
            # Shape: (batch_size, n-k)
            dot_products_xj = torch.gather(dot_products_all, 1, self.
                subset_excluded_tensor)
            # Calculate differences and apply ReLU
            # We need to unsqueeze dot_products_xi to match dimensions for
                broadcasting
            # Shape: (batch_size, n-k) - (batch_size, k)
            differences = dot_products_xi.unsqueeze(1) - dot_products_xj.
                unsqueeze(2)
            total_loss = torch.relu(-differences).sum((-1, -2)).mean()
            num_violations = (differences < 0).sum().item()
            return total_loss, num_violations

    def train(self, d, num_epochs, learning_rate=0.1, patience=20):
        self.vector_embeddings = torch.randn(
            self.n, d, device=self.device, requires_grad=True
        )

        optimizer = optim.Adam(
            [self.vector_embeddings],
            lr=learning_rate,
        )
        scheduler = optim.lr_scheduler.OneCycleLR(
            optimizer=optimizer,
            max_lr=learning_rate,
            total_steps=num_epochs,
            pct_start=0.0,
        )

        min_violations = self.total_violations
        epochs_no_improve = 0

        with trange(num_epochs, desc=f"\t\t_n={self.n},_k={self.k},_{d=}"
            ) as epoch_iterator:
            for epoch in epoch_iterator:
                optimizer.zero_grad()

                loss, violations = self.calculate_loss()
                loss.backward()
                optimizer.step()
                scheduler.step()

                epoch_loss = loss.item()
                epoch_iterator.set_postfix(
                    {
                        "n": self.n,
                        "k": self.k,
                        "loss": epoch_loss,
                        "#vio_rate": violations / self.total_violations,
                        "min_#vio": min_violations,
                        "epochs_no_improve": epochs_no_improve
                    }
                )

                if violations < min_violations:
```

```python
                    min_violations = violations
                    epochs_no_improve = 0
                else:
                    epochs_no_improve += 1

                if violations == 0:
                    print("Early_stopping:_No_violations_found.")
                    break
                if epochs_no_improve >= patience:
                    print(f"Early_stopping:_No_improvement_in_violations_
                        for_{patience}_epochs.")
                    break

        return min_violations

class Experiment:
    def __init__(self):
        self.search_paths = {}
        self.minimal_dimensions = {}

    def find_minimal_dimension(self, k, n_values, left0=0, num_epochs
        =100, learning_rate=0.1, patience=10):

        last_minimal = left0
        for n in n_values:
            print("#" * 10 + "new_task" + "#" * 10)
            print(f"Finding_minimal_dimension_for_n={n},_k={k}")
            print("#" * 30)
            trainer = Trainer(n, k)

            self.search_paths[n] = []
            left, right = last_minimal+1, last_minimal + 40 # the range
                is set by prior to accelerate the convergence.
            minimal_d = n + 1

            while left <= right:
                mid = (left + right) // 2
                if mid == 0:
                    mid = 1

                print(f"\t>>>>Testing_dimension_d={mid}")

                violations = trainer.train(
                    d=mid,
                    num_epochs=num_epochs,
                    learning_rate=learning_rate / np.log2(n),
                    patience=patience
                )

                print(f"\t<<<<Violations_for_d={mid}:_{violations}")

                self.search_paths[n].append({'dimension': mid, '
                    violations': violations})

                if violations == 0:
                    minimal_d = mid
                    right = mid - 1
                else:
                    left = mid + 1

            self.minimal_dimensions[n] = minimal_d if minimal_d <= n else
                -1
            last_minimal = minimal_d
```

```python
                with open("minimal_dem_log.txt", "at") as f:
                    f.write(f"minimal_dimension_@_{k=}&{n=}_is_{minimal_d}\n"
                        )
            print("#" * 10 + "_Task_Finished" + "#" * 10)
            print(f"minimal_dimension_@_{k=}&{n=}_is_{minimal_d}\n")
            print("#" * 30)

        return self.minimal_dimensions

    def plot_minimal_dimension_vs_n(self, k):
        if not self.minimal_dimensions:
            print("No_experiment_results_to_plot._Run_
                find_minimal_dimension_first.")
            return

        n_plot = list(self.minimal_dimensions.keys())
        d_plot = list(self.minimal_dimensions.values())

        plt.figure(figsize=(8, 6))
        plt.plot(n_plot, d_plot, marker='o', linestyle='-')
        plt.xlabel("Number_of_Vectors_(n)")
        plt.ylabel("Minimal_Dimension_(d)")
        plt.xscale('log')
        plt.title(f"Minimal_Dimension_vs._Number_of_Vectors_for_k={k}")
        plt.grid(True)
        plt.show()

        plt.savefig("med_vs_n.png")

    def plot_violations_vs_d(self, k):
        if not self.search_paths:
            print("No_experiment_results_to_plot._Run_
                find_minimal_dimension_first.")
            return

        for n, path in self.search_paths.items():
            sorted_path = sorted(path, key=lambda x: x['dimension'])

            dimensions = [item['dimension'] for item in sorted_path]
            violations = [item['violations'] for item in sorted_path]

            plt.figure(figsize=(8, 6))
            plt.plot(dimensions, violations, marker='o', linestyle='-')
            plt.xlabel("Dimension_(d)")
            plt.ylabel("Number_of_Violations")
            plt.title(f"Violations_vs._Dimension_for_n={n},_k={k}")
            plt.grid(True)
            plt.show()

# 1. Define a list of n_values to experiment with
n_values = [5 * 2 ** (i) for i in [0, 1, 2, 3, 4, 5]]

# 2. Set a value for k
k = 2

# 3. Instantiate the Experiment class
experiment = Experiment()

# 4. Call the find_minimal_dimension method with a reasonable batch_size
#    and limited epochs
#    Using a batch_size of 32 and num_epochs of 100 for initial testing.
print("Starting_experiment_with_mini-batching_and_early_stopping...")
minimal_dims = experiment.find_minimal_dimension(
```

```
213        k, n_values, learning_rate=1, num_epochs=1000, patience=1000)
214
215    # 5. Print the minimal_dims dictionary
216    print("\n_Minimal_dimensions_found:", minimal_dims)
217
218    # 6. Call the plot_minimal_dimension_vs_n method
219    experiment.plot_minimal_dimension_vs_n(k)
220
221    # 7. Call the plot_violations_vs_d method
222    experiment.plot_violations_vs_d(k)
```

