# OpenReview forum: "Tight Bounds and Achievable Upper Bounds of Minimal Dimensions for Embedding-based Retrieval"
_ICLR.cc/2026/Conference — Submitted to ICLR 2026_

### Official Review · Reviewer_Dok3 · 2025-10-27

**Soundness:** 3
**Presentation:** 1
**Contribution:** 2
**Rating:** 4
**Confidence:** 4

**Summary:**

The paper provides theoretical bounds for the so-called minimal embeddable dimension, which is the smallest dimension for which some configuration of m points with a given functional family can be k-shattered. It shows that both lower and upper bounds are independent of the number of points, and only depend on k, in the special case where the functional family is given by the 3 standard scoring functions: inner product, cosine, and L2 distance.

The paper also defines so-called minimal achievable embeddable dimension, where k-shattering is replaced by k achievable-shattering, defined by evaluating a scoring function evaluated at the centroid of k nearest points. Here nearest just means highest scores. Then the paper uses a union bound to show that the O(k^2 log m) dimension is sufficient to find a configuration of m points with the k achievable-shattering property. Finally the paper runs a simulation to verify that the true relation between achievable configuration and dimension is indeed logarithmic as opposed to cubic in a referenced paper.

**Strengths:**

The context of the problem being addressed is of significant interest in the ML community. For instance, in K nearest neighbor retrieval, we need to find the right kind of dimension to ensure most if not all query embeddings can find its nearest k item embeddings simply using dot product, L2 distance, or cosine distance (dot product with item embeddings L2-normalized).

The construction using the moment curve is interesting mathematically.
The proof of the achievable upper bound is also standard and reasonable. The simulation result also supports its general order of magnitude.

**Weaknesses:**

The exposition of the paper is quite cryptic sometimes. I will list some examples
l070-075: looks like 3 things are being compared here: MED, MAED, and real life practical situation. The first sentence says MAED is weaker than real life, but the second sentence then concludes that MAED upper bounds MED. The logic simply doesn’t follow.
It would be helpful to tabulate the results for the 3 kinds of scoring functions to make their relationship more transparent.

There are many typos in the paper, including
l293: “We use optimize m embeddings randomly initialized”
l028-029: "retrieving the top-k answers of top-k largest scores" should be rephrased as "retrieving the answers with the k largest scores".

l019 (abstract): "Our results also align well with existing practices in large language models, vector databases, and other related fields." seem irrelevant.

l097: C_k definition doesn't need min

The paper can also make the result statements as well as the proofs more accessible to general ML practitioners not familiar with proof heavy literature, involving VC dimensions.

The upper bound result of the MED is pretty contrived. It doesn’t really show that the dimension n doesn’t depend on the number of points, but only that for any m, one can find a configuration of m points in R^n with the property that the k nearest points of any point can be separated by one of the scoring functions. This makes the result of little practical value.

Even in the MAED case, the so-called achievable query, where it’s required to be the centroids of its k nearest neighbors, seems rather special. In addition, it’s again not saying the dimension upper bound works for all configurations, but rather one can find some configuration with the achievability constraint under the upper bound, even though it should work for almost all cases. I think it’s very much worth highlighting the limitations of these results, and try to make some attempt connecting the results to practical cases, such as in a unsupervised KNN learning task.

The setup for the simulation requires more detailed explanation as well as motivation. The use of gradient descent to search for an achievable k-shattering configuration does not guarantee optimality, but is sufficient for the upper bound. Some references would be useful to compare against other such simulation work for such a theoretical result.

**Questions:**

Overall I think the paper has some interesting probabilistic results. But it needs to explain the results in a more accessible manner. There should be plenty of space left to add more details in the main text.
I would like to see the paper much better polished in terms of writing style and motivation. Focus on the main claims, namely the 2 upper bounds and 1 lower bound, as well as the special treatment for each scoring function, and making sure the setup, definitions, and proof strategy are completely transparent. Leave some of the propositions to the appendix.

---

> ### Author Response · Authors · 2025-11-26
>
> Dear reviewer Dok3
>
> Thanks for your numerous kind suggestions that helped substantially improve the quality of our work.
>
> We fully agree with you that our results focus only on approximability in embedding-based retrieval, rather than on the learnability side.
>
> All comments about the presentation have been addressed in the revision.
>
> In addition, we have made the following modifications to make the paper more focused, coherent, and scientific:
> 1. Restate the goal of this paper and make a clear separation of approximability and learnability. Based on that, it highlights the scope and limitations of each bound.
> 2. Refine the writing with sufficient details to enrich the context for all essential claims. Add backgrounds to the proof-intensive contents, such as VC dimension.
> 3. Reconnect the paper with the central claims of previous works. More discussions are being held to clarify the message.
> Thanks again for your comments and time.
>
> Thanks again for your guidance.
>
> Best regards

---

### Official Review · Reviewer_KFqx · 2025-10-29

**Soundness:** 3
**Presentation:** 1
**Contribution:** 2
**Rating:** 2
**Confidence:** 4

**Summary:**

This work studies the *minimal embeddable dimension* (MED) problem, where given a set of $m$ objects
and a pairwise scoring function $f$, we want to know the minimum embedding dimension $n$ such that
we can perfectly recover the top-k (object, query) results according to $f$. The authors consider
the following score functions: Euclidean distance, cosine similarity, and a related inner product.
They also study a so-called "achievable" setting (MAED) where query vectors are of the form
$1/|S| \sum_{i \in S} \mathbf{x}_i$ for some set $S \subseteq X$ of size $k$. They give a clean
proof of the lower and upper bounds for MED (tight up to a factor of 2). They also give
$O(k^2 \log m)$ upper bounds for MAED and some synthetic experiments to demonstrate this result.

**Strengths:**

- The authors introduce the study of the achievable setting.
- The MED and MAED problem statements and analysis are built on a clean definition of *$k$-shattering*.
- The cyclic polytope example in Section 3.1 is instructive and succinct.

**Weaknesses:**

- Manuscript is quite unpolished.
- Experiments (Section 4.2) are very interesting but incomplete. It would be
  good in a future version of this paper to strengthen these results (e.g.,
  revisiting the cyclic polytope as a warm up).

**Questions:**

**Questions**

- What are the lower bounds for MAED (Table 1)?

**Misc**

- [049] Typo: "can be found in Section ??"
- [081] Typo: two different capitalization styles for list items, i.e., "Standard setting" and "achievable setting"
- [081] Typo: "simulation results on the ."
- [095] Nit: How do we handle having the same vector for two different items given the notation $\{\mathbf{x}_{i}\}_{i=1}^m$?
- [102] Nit: Inconsistent use of normal and boldface letters for scalars and vectors.
- [246] Typo: "top-k" --> "top-$k$"
- [281] Sugestion: Add some horizontal space between the captions of Figure 1 and Figure 2 so it's more clear they're separate.

---

> ### Author Response · Authors · 2025-11-26
>
> Dear reviewer KFqx,
>
> Thanks for your critical comments and suggestions to subtantially improved this paper.
>
> > Manuscript is quite unpolished.
>
> Thanks for your suggestions regarding the presentation issues. We have polished the paper thoroughly. Please check the details.
>
> > Experiments (Section 4.2) are very interesting but incomplete. It would be good in a future version of this paper to strengthen these results (e.g., revisiting the cyclic polytope as a warm up).
>
> We sincerely thank the reviewers for these suggestions and have polished the paper to make it more accessible. However, the cyclic polytope as a warm-up may lead to other risks, such as numerical precision, so that they can not be used as a universal way of warm-up. We discuss the issues of cyclic polytope in the revised version (see paragraph Can the $\Theta(k)$ bounds be easily observed?), and that could explain why we need a centroid version for numerical simulation.
>
> > What are the lower bounds for MAED (Table 1)?
>
> The lower bound of MED is a natural lower bound of MED-C (renamed from the old MAED). For any configuration embeddable in a MED-C setting, one can just use the subquery embeddings as the centroid of the embeddings of the elements they contain. In this way, we construct a MED configuration. That means, we always have MED ≤ MED-C; therefore, the lower bound of MED is the lower bound of MED-C. We have modified the paper thoroughly.
>
> > [049] Typo: "can be found in Section ??"
> > [081] Typo: two different capitalization styles for list items, i.e., "Standard setting" and "achievable setting"
> > [081] Typo: "simulation results on the ."
> > [095] Nit: How do we handle having the same vector for two different items given the notation \({\mathbf{x}{i}}{i=1}^m\)?
> > [102] Nit: Inconsistent use of normal and boldface letters for scalars and vectors.
> > [246] Typo: "top-k" --> ...
> > [281] Sugestion: Add some horizontal space between the captions of Figure 1 and Figure 2 so it's more clear they're separate.
>
> These have been addressed in the updated manuscript.
>
> Best regards

---

### Official Review · Reviewer_txVp · 2025-10-31

**Soundness:** 2
**Presentation:** 4
**Contribution:** 3
**Rating:** 6
**Confidence:** 4

**Summary:**

This paper investigates the minimum dimension of the vector space required for embedded retrieval systems, aiming to challenge the view on the "vector bottleneck" in the field. The authors introduce two core settings to analyze this problem:

1. Standard Setting. Under this theoretically idealized setting, the paper proves that the Minimum Embedding Dimension (MED)—required to perfectly retrieve all queries with no more than k answers—has a linear relationship only with $k$ (i.e., $\Theta(k)$) and is independent of the total number of objects $m$ in the corpus.
2. Achievable Setting. Under this more practically relevant setting, query vectors are constrained to be the centroid of the answer set vectors. The paper theoretically derives and experimentally verifies that the Minimum Achievable Embedding Dimension (MAED) required for this constructive method has a logarithmic relationship with the total number of objects $m$ (i.e., $d=\text{O}(k^2\text{log}m)$).

**Strengths:**

The theoretical proof on the $\Theta(k)$ bound for the Minimum Embedding Dimension provides an entirely new, more optimistic perspective for understanding the theoretical limits of embedded retrieval. On top of that, the paper successfully reframes the "vector bottleneck" problem, shifting it from a seemingly immutable hardware constraint (space dimension) to an optimizable software issue (embedding construction method).

**Weaknesses:**

1. The experimental validation for the "achievable" $O(\log m)$ bound is not truly achievable, as its training method requires checking all $\binom{m}{k}$ combinations, which is computationally unfeasible for large $m$.

2. The experimental comparison to prior work [Weller et al., 2025a] is misleading because it compares results from two different paradigms (MAED vs. MED). A fair comparison would require the authors to re-run experiments under the same (e.g., MED) setting to prove their optimization method is truly better.

3. The centroid query method proposed in the paper cannot handle complex compositional queries, which limits the applicability of the achievable method. It is important to more honestly define the scope of applicability of their method.

4. There are several minor but noticeable typographical and notational issues in the manuscript.

**Questions:**

1. Most importantly, to truly support the "achievable" claim, you should demonstrate that this O(logm) bound can also be reached using a scalable, practical training algorithm (e.g., one based on negative sampling or contrastive loss) rather than the full-combination check.

2. To make a more robust claim, you should run your optimization method under the same MED (Standard Setting) as Weller et al. This would provide a true "apples-to-apples" comparison and prove that the difference is due to your superior optimization, not the change in settings. Alternatively, provide a clear theoretical proof or new experimental evidence within your paper demonstrating that the MAED (centroid query) is indeed a harder problem than MED (free query).

3. You should more clearly define the limited scope of the MAED model. It would be valuable to discuss the gap between this "centroid query" model and a more realistic "independent query" model, and what new challenges the latter might introduce.

4. There are several minor but noticeable typographical and notational issues in the manuscript. It is recommended that the authors carefully proofread the paper to improve readability and consistency. In particular:
    - There are some cross-reference errors, including "Section ??"  and "on the \space(a space)" in Section 1.
    - Some misspellings are present, such as "MEAD" in Section 2.3 and "k-shuttering" in Section 3.2.
    - Equation 10 in A.2 should be $\langle v_1, \sum_{u\in S} u\rangle - \langle v_2, \sum_{u\in S} u\rangle$.

---

> ### Author Response · Authors · 2025-11-26
>
> Dear reviewer txVp,
>
> Thank you for your suggestion, which helped us substantially improve this manuscript.
>
> We are sorry that there are some misunderstandings regarding the meaning of “achievable”. There are many perspectives on “achievable,” and we agree with you that our numerical simulation requires optimization over the full dataset, which, as you pointed out, cannot be realized in a large, practical dataset.
>
> However, we may not have made the key point of our paper clear enough to you. And we find that many of the weaknesses you have raised stem from misunderstandings. We would like to emphasize that:
>
> 1. The central goal of this paper is to find bounds on the minimal embeddable dimension, which are really about the space's internal capacity. In other words, the MED identifies the minimal dimension of a space in which an embedding-based retrieval model can exist, but it ignores how to find such a model. MED addresses the embeddability problem from an approximability perspective, rather than a learnability perspective. The bounds of MED hold a position similar to that of the VC dimension for classifiers and to the universal approximability of MLP networks, both of which are milestone results in the development of classic ML methods.
>
> 2. Based on 1, our numerical simulation in the MED-C (renamed from the confusing MAED) setting is NOT intended to propose a model (so we are sorry you pointed out these model issues in weakness #3), but to estimate an upper bound on the MED in an empirical, achievable way. Several critical points should be made clear.
>     1. All empirical estimates of the MED are upper bounds. Because if one finds a configuration of element embeddings and subset query embeddings that are embeddable in a particular dimension $d$, the MED is no larger than $d$ by definition. Empirical estimation is important because it at least tests the correctness of the theory.
>     2. An achievable way to estimate such an upper bound is also essential. Our MED-C, or centroid setting in the revised manuscript, is “more achievable” compared to the “free-embedding optimization” method proposed in Weller et al. In “free-embedding optimization”, one estimates the upper bound of minimal embeddable dimension given $m$ points (or the lower bound of the maximal number of embeddable points given dimension $d$) by optimizing $m$ element embeddings and ${m \choose k}$ subset query embeddings. It contains a combinatorially large number of variables. In the MAED setting, we only optimize $m$ element embeddings and leave the other ${m\choose k}$ to use the centroid of their contained answers. We agree with you that this aggressive simplification failed to capture the modern capability of neural networks, but, in another way, can also upper-bound MED.
>
> Please check the revised version to find how those points are made in a broader context.
>
> Responses to the specific weaknesses are listed in the next post.

---

> > ### Author Response · Authors · 2025-11-26
> > **Rebuttal part 2**
> >
> > > The experimental validation for the "achievable" bound is not truly achievable, as its training method requires checking all combinations, which is computationally unfeasible for large m.
> > > Most importantly, to truly support the "achievable" claim, you should demonstrate that this O(log m) bound can also be reached using a scalable, practical training algorithm (e.g., one based on negative sampling or contrastive loss) rather than the full-combination check.
> >
> > We fully agree with this. But studying the learnability is not our goal. We focus on approximability rather than learnability. Please check point 1 above.
> >
> > > The experimental comparison to prior work [Weller et al., 2025a] is misleading because it compares results from two different paradigms (MAED vs. MED). A fair comparison would require the authors to re-run experiments under the same (e.g., MED) setting to prove their optimization method is truly better.
> > > To make a more robust claim, you should run your optimization method under the same MED (Standard Setting) as Weller et al. This would provide a true "apples-to-apples" comparison and prove that the difference is due to your superior optimization, not the change in settings. Alternatively, provide a clear theoretical proof or new experimental evidence within your paper demonstrating that the MAED (centroid query) is indeed a harder problem than MED (free query).
> >
> > We disagree with this. Again, this paper does not aim to provide any new method or datasets, but only focuses on the theoretical perspective. “Free embedding optimization” in Weller et al. is used to empirically estimate the minimal embeddable dimension for $m$ points (or the lower bound on the maximal number of embeddable points given dimension $d$). The MED-C setting also aims to achieve the same goal. In the MED-C setting, far fewer parameters are optimized than in Weller et al.. Yet, they reveal logarithmic growth, which is significantly slower than the polynomial growth reported in Weller et al. We can then claim that the MED-C and MED should grow no faster than $O(\log m)$, which agrees with the theoretical bounds.
> > The empirical evidence supports the $O(\log m)$, which suggests that the dimension required actually grows slowly with the number of points in a retrieval system. This contradicts the central claims in Weller et al. that “we may encounter the theoretical limits (on vectors in geometric space)”. The $O(\log m)$ is NOT the theoretical limit to blame. We want to send a message to the community that there is no limit in approximability, but it still lies in learnability.
> >
> > > The centroid query method proposed in the paper cannot handle complex compositional queries, which limits the applicability of the achievable method. It is important to more honestly define the scope of applicability of their method.
> > > You should more clearly define the limited scope of the MAED model. It would be valuable to discuss the gap between this "centroid query" model and a more realistic "independent query" model, and what new challenges the latter might introduce.
> >
> > We fully agree with you that the centroid setting is too simple to handle anything like a complex query. But let’s focus on the primary goal of this paper, which is simple but enough to show the $O(\log m)$ empirical observation. It is sufficient to contradict previous works. And, we also discuss the situation
> >
> > > There are several minor but noticeable typographical and notational issues in the manuscript. It is recommended that the authors carefully proofread the paper to improve readability and consistency. In particular:
> > > ○ There are some cross-reference errors, including "Section ??" and "on the \space(a space)" in Section 1.
> > > ○ Some misspellings are present, such as "MEAD" in Section 2.3 and "k-shuttering" in Section 3.2.
> > > ○ Equation 10 in A.2 should ...
> >
> > We sincerely thank the reviewers for these suggestions, and we have polished the paper to make it more accessible.
> >
> > Best regards,

---

### Official Review · Reviewer_JMqo · 2025-10-31

**Soundness:** 4
**Presentation:** 3
**Contribution:** 4
**Rating:** 8
**Confidence:** 4

**Summary:**

The paper studies the problem of finding the appropriate dimensionality to embed data in vector spaces. In contrast with recently published work, the formal findings in this paper show an encouraging picture for embedding based retrieval. First, for common similarity measures, the minimal number of dimensions does not depend on the cardinality of the set to embed. Second the minimal dimensionality is a low degree polynomial of the number k of retrieved vectors: between k and 2k in the general case and quadratic in the "achievable" setting. Considering that the number of retrieved vectors is a small number in practice, these theoretical bounds paint a positive picture for vector search.

**Strengths:**

- The topic is of great relevance in practice, even though the results are of a very theoretical nature.
- The dimensionality bounds are novel and they bring a much needed formal understanding to the important area of retrieving unstructured data.
- These small bounds also highlight that more work is needed in the embedding models and that such practical work is not a lost cause.
- To the best of my knowledge, the proofs are correct and the level of rigor exhibited is appropriate for ICLR.

**Weaknesses:**

- The discussion of the achievable setting in the introduction (line 70) feels a bit lacking and its description in the contributions (lines 81 to 84) too vague. The authors should position the achievable setting more clearly in a "hardness of embbedability" scale.
- How tight are the MAED bounds? Although the authors state that this bound may not be tight, I would have appreciated a more detailed discussion of what this bound means in practice.
- How important is the fact that random vectors are used to get the MAED bound? What would happen in a different setting? Would the bound get worse? It is important to clearly state whether the authors are covering a best or worst case scenario here (or neither and it is just a particular one). This should be more clearly stated in the introduction and abstract, because those section seem to implicitly indicate the MAED bound is general.

**Questions:**

- In the abstract, it is unclear what the authors mean by achievable. It would be useful to have a succint definition.
- In the third paragraph  of the introduction, too little context is given when taking about the work by Weller et al. It is not clear what the query-relevance matrix and rank_{+/-} mean. Adding a couple of sentences might help. Alternatively, the authors should up-level that discussion to a more intuitive explanation.
- There is an undefined symbol at the end of the third paragraph (line 49).
- In the contributions, there is an unfinished sentence in the item about the "achievable setting" (line 81).
- After Definition 2.11, "that if MAED" -> "that MAED" (line 181).
- After Proposition 2.12, "MEAD" -> "MAED" (lines 186 and 187)
- Please add a horizontal space between Figures 1 and 2 as the captions are hard to read.

---

> ### Author Response · Authors · 2025-11-26
>
> Dear reviewer JMqo,
>
> Thanks again for your valuable comments that substantially improved the quality of this paper.
>
> > The discussion of the achievable setting in the introduction (line 70) feels a bit lacking and its description in the contributions (lines 81 to 84) too vague. The authors should position the achievable setting more clearly in a "hardness of embbedability" scale.
>
> We really appreciate your suggestions in making this paper more accessible. Please check our updates in the introduction section.
>
> > How tight are the MAED bounds? Although the authors state that this bound may not be tight, I would have appreciated a more detailed discussion of what this bound means in practice.
>
> The tightness of MED-C bounds (renamed from the old MAED) can be understood by comparing them to their MED $\Theta(k)$ bound. We want to emphasize that MED-C cannot be lower than MED (if there is such a configuration of element embeddings to achieve MED-C, one can just find out proper subset embeddings by centroids to make this configuration embeddable, and MED is smaller than that by definition). Current upper bound of MED-C $O(k^2\log(m))$ differs from MED $\Theta(k)$ by a factor of $k\log(m)$, which is logarithmic to $m$ and linear to $k$. We can usually say it is not a very large gap in $\log(m)$, but still some gap in \(k\). However, theoretical research on related problems suggests that further reducing the upper bound to $O(k\log m)$ might incur additional computational cost, such as in sparse recovery.
>
> More discussions can be found in the new paragraph in the revised manuscript.
>
> > How important is the fact that random vectors are used to get the MAED bound? What would happen in a different setting? Would the bound get worse? It is important to clearly state whether the authors are covering a best or worst case scenario here (or neither and it is just a particular one). This should be more clearly stated in the introduction and abstract, because those section seem to implicitly indicate the MAED bound is general.
>
> The random vector approach, or probabilistic method in general, is just an approach to prove the existence of a configuration of element embeddings, such that, when taking the subset query embedding as the centroid of its answers, it is embeddable by showing its probability is larger than zero. This existential proof matches your word “best case scenario” and is suitable here because we are considering the minimal (centroid) embeddable dimension. We are dedicated to making our presentation as clear as possible. Please kindly review the updated manuscript.
>
> > In the abstract, it is unclear what the authors mean by achievable. It would be useful to have a succint definition.
> > In the third paragraph of the introduction, too little context is given when taking about the work by Weller et al. It is not clear what the query-relevance matrix and rank_{+/-} mean. Adding a couple of sentences might help. Alternatively, the authors should up-level that discussion to a more intuitive explanation.
> > There is an undefined symbol at the end of the third paragraph (line 49).
> > In the contributions, there is an unfinished sentence in the item about the "achievable setting" (line 81).
> > After Definition 2.11, "that if MAED" -> "that MAED" (line 181).
> > After Proposition 2.12, "MEAD" -> "MAED" (lines 186 and 187)
> > Please add a horizontal space between Figures 1 and 2 as the captions are hard to read.
>
> Other typos, suggestions, and comments about the presentation are addressed in the newest version.
>
> Best regards,

---

### Official Review · Reviewer_KFwt · 2025-11-01

**Soundness:** 3
**Presentation:** 2
**Contribution:** 4
**Rating:** 6
**Confidence:** 4

**Summary:**

This submission looks at the problem of determining the minimum dimension needed to encode an arbitrary set $X$ of $m = |X|$ objects into $\mathbb{R}^n$ such that a retrieval query on $X$ with $k$ answers in the set is perfectly retrievable. This is done by way of a combinatorial argument:
1. $k$-shattering is used to define the minimum embeddable dimension (MED)
2. A simple VC-dimension bound falls out of the definitions, parametrized by $m$ and $k$, and works for an arbitrary scoring function "family" $\mathcal{F}$
3. For specific scoring functions (dot products, $\ell_2$, cosine sim.), one can get tight bounds on the MED of $\Theta(k)$
4. A minimum achievable dimension (MAED) is designed to model practical scenarios and provides an upper bound on MED
There is also an upshot given by empirical results, which suggest that how we construct the embeddings (or generate them with a neural net) matters much more than the available dimensions.

**Strengths:**

1. The bounds on MED are quite surprising and make for a great result.
2. The contrasting optimism for low-dimensional dense retrieval to the prior work of Weller, et al. will make for interesting and important discussion on the limits of vector search in the AI landscape.
3. Careful effort is made to reconcile empirical results with the theoretical results in an intuitive manner. There is also a clean comparison with the prior work, which makes it easier to reconcile the position of this work with existing results.

**Weaknesses:**

1. Typos (e.g. lines 49, 81, 283): some are quite substantial, definitely get these fixed
2. Considering that this submission aims to contradict earlier work, some more discussion about the earlier work, what is acking in it, and motivation to pursue this approach in place of the prior work should appear earlier on in the manuscript.
3. The MAED discussion is perhaps oversimplifying the practical scenarios that it tries to represent. While it does model the in-distribution setting of vector search, it fails (at the admission of the authors) to capture the nuance that comes with embeddings generating with a neural network. This leads to an empirical section that is, I feel, lacking. To complement the existing results, there should be experiments based on real data with neural network based embeddings that support the paper's results and an effort to quantify the issues that come with such a setting.

**Questions:**

1. It's common in practice to retrieve a larger number (than $k$) of candidates, then rerank them down using a stronger similarity function into $k$ final results. Is there a way to model that setting with this framework? If we were to naively apply the theoretical results to this method, it's possible we would see poor results (as we rely on $k << m$), but this seems to work remarkably well in practice.

---

> ### Author Response · Authors · 2025-11-26
>
> Dear reviewer KFwt,
>
> Thanks for your valuable feedback.
>
> > Typos (e.g. lines 49, 81, 283): some are quite substantial, definitely get these fixed
>
> All typos have been fixed, and the paper has undergone a thorough proofreading.
>
> > Considering that this submission aims to contradict earlier work, some more discussion about the earlier work, what is lacking in it, and motivation to pursue this approach in place of the prior work should appear earlier on in the manuscript.
>
> We have added another paragraph in both the discussion and introduction sections to highlight the differences between our bounds and the previous work. In short, the key differences are twofold:
> 1. We establish the explicit relation of the minimal dimension required with respect to the number of elements $m$ and the largest cardinality of the set $k$, which was deferred to the computation of signed rank in previous work.
> 2. Our bounds suggested that one of the key evidence – “number of top-$k$ subsets of documents capable of being returned as the result of some query is limited by the dimension of the embedding” which is indirectly suggested by their theory and directly suggested from their experiments - may not be correct.
>
>
> > The MAED discussion is perhaps oversimplifying the practical scenarios that it tries to represent. While it does model the in-distribution setting of vector search, it fails (at the admission of the authors) to capture the nuance that comes with embeddings generating with a neural network. This leads to an empirical section that is, I feel, lacking. To complement the existing results, there should be experiments based on real data with neural network based embeddings that support the paper's results and an effort to quantify the issues that come with such a setting.
>
> Thanks for raising this issue. The discussion about MED-C (changing from the MAED naming) is not intended to be at a practical level of experimentation. The primary focus of MED-C simulations is twofold:
> 1. We show that our bound is correct in terms of $\log(m)$.
> 2. Compared with the free embedding optimization conducted by Weller et al., a simpler setting (as elaborated in the new revision) can find configurations of element embeddings and subset embeddings in much lower dimensions.
>
> > It's common in practice to retrieve a larger number (than $k$) of candidates, then rerank them down using a stronger similarity function into $k$ final results. Is there a way to model that setting with this framework? If we were to naively apply the theoretical results to this method, it's possible we would see poor results (as we rely on $k << m$), but this seems to work remarkably well in practice.
>
> Thanks for mentioning this practical setting. One possible explanation lies in the hierarchical nature of real-world data. The assumption of data hierarchy, although we cannot prove it here, is supported by many empirical successes, such as clustering and hyperbolic representation learning. Now let’s analyze how the MED and MED-C upper bounds apply to a toy hierarchical data, say, $m$ clusters and $m$ elements in each of the clusters.
> 1. For MED bounds, it does not require $k \ll m$; it is just a fact of combinatorial geometry. We put $m^2$ points into the cyclic polytope, and that's it.
> 2. To apply MED-C bounds, suppose we want to find the top-$k$ relevant elements, but into a two-step approach: firstly, we find top-$c$ clusters of interest (all elements in those clusters are the results of the first round of filtering) and then find top $a k/c$ elements in each of the clusters, where $a>1$ is a constant factor to capture more points in the search result. In the first step, we need $d_1 = O(c^2 \log (m))$ dimension and in the second step, we need $d_2 = O(a^2k^2/c^2 \log (m))$ dimension. Suppose we concatenate embeddings for two steps together, resulting in $O((c^2 + a^2k^2 / c^2)\log(m))$ dimension. The practical scenario can be viewed to evaluate the first similarity function on the first $d_1$ dimensions and then the last $d_2$ dimension. The “best case” analysis here suggests that we can hopefully finish this procedure in $O((c^2 + a^2k^2 / c^2)\log(m)) \geq O(2ak \log(m))$ case, which reduces the best case form $O(k^2 \log (m))$ to $O(k \log (m))$.
>
> Thanks again for your valuable comments that substantially improved the quality of this paper.
>
> Best regards,
> authors.

---

### Author Response · Authors · 2025-11-26
**Response to All Reviewers**

Dear Reviewers,

We sincerely thank you for your time, effort, and constructive feedback. Your insights have greatly improved the manuscript.

## Summary of Updates
- **Polishing and Clarity**: Fixed all typos, notation inconsistencies, cross-references, and unfinished sentences. Added background on concepts like VC-dimension for accessibility. Improved figure spacing and rephrased awkward sections.
- **Renaming and Restructuring**: Renamed "achievable setting" (MAED) to "centroid setting" (MED-C). Updated Table 1 to clarify MED ≤ MED-C, omitting redundant $\Omega(k)$ lower bound for MED-C.
- **Enhanced Discussions**: Added paragraphs in introduction and discussion comparing our $\Theta(k)$ bounds (m-independent) to prior work (e.g., Weller et al., 2025a), resolving $m, k$ dependencies and contrasting logarithmic vs. polynomial empirical growth.
- **Scope and Limitations**: Clarified focus on approximability (geometric constraints) vs. learnability (training challenges). Discussed centroid setting limitations (e.g., no complex queries) and practical implications.

## Addressing Common Concerns
- **Typos, Presentation, and Accessibility** (KFwt, JMqo, txVp, KFqx, Dok3): Manuscript fully polished; added intuitive explanations for definitions, proofs, and prior work (e.g., query-relevance matrices).
- **Approximability vs. Learnability** (txVp, Dok3): Emphasized results prove existence in geometry, not universal training; centroid is constrained but counters prior polynomial claims with O(log m) evidence.
- **Tightness and Empirical Validation** (JMqo, KFqx): MED-C upper bound O(k² log m) has k log m gap to MED Θ(k); simulations validate O(log m) via gradient descent for upper-bound existence.
- **Comparison to Prior Work** (KFwt): Highlighted explicit m/k bounds over sign-rank; suggests limits are in learnability, not geometry.

We look forward to your thoughts.

Best regards,
The Authors

---

### Author Response · Authors · 2025-12-03
**Summary of the rebuttal period.**

Dear Area Chair,

We sincerely thank you for your time on our submission. Here are some key summaries about the rebuttal period.

### Summary of Updates

In response to reviewers' feedback, we have revised the manuscript as follows:

- **Polishing and Clarity**: We have thoroughly proofread the paper to fix all identified typos, inconsistencies in notation, and errors. We have also improved the exposition for better accessibility, including background explanations of technical concepts such as VC-dimension, to make the proofs more accessible to general ML practitioners. Horizontal spacing has been added between figure captions for readability.

- **Renaming and Restructuring**: The "achievable setting" (previously MAED) has been renamed to the "centroid setting" (MED-C) to more accurately reflect its focus on subset embeddings as centroids of element embeddings. This change emphasizes its role as a more constrained, yet empirically verifiable, configuration. We have updated Table 1 to clarify that MED's lower bound naturally applies to MED-C (since any MED-C configuration can be converted to a MED one by deriving subset embeddings via centroids, ensuring MED ≤ MED-C). We have also omitted the redundant $\Omega(k)$ lower bound for MED-C in the table for conciseness.

- **Enhanced Discussions**: We have added paragraphs in the introduction and discussion sections to better position our work relative to prior studies (e.g., Weller et al., 2025a). This includes explicit comparisons of our tight bounds ($\Theta(k)$ for MED, independent of $m$) with their sign-rank approach, highlighting how we resolve the dependency on $m$ and $k$ that was previously deferred. We have also elaborated on the empirical simulations, reconciling them with theoretical results and contrasting the logarithmic growth in our centroid setting with the polynomial growth in prior free-embedding optimizations.

- **Scope and Limitations**: We have expanded sections to clearly delineate the approximability focus of our work (geometric constraints) from learnability challenges (e.g., training neural networks for embeddings). This includes discussions on the limitations of the centroid setting (e.g., its simplicity may not handle complex compositional queries) and the contrived nature of some upper bounds (e.g., they prove existence for specific configurations rather than universality). We have added context on practical implications, such as alignment with vector databases and large language models, while noting gaps in real-world scenarios, such as reranking or hierarchical data.

### Common Concerns are All Addressed.

- **Distinction Between Approximability and Learnability**: A key theme across reviews (e.g., txVp, Dok3) is the practical value of our bounds and the limitations of the centroid setting. This is a misunderstanding. We emphasize that our results focus on approximability—the fundamental geometric capacity of vector spaces to embed subset memberships—rather than learnability (e.g., scalable training with neural networks or negative sampling). The MED upper bound ($O(k)$) shows that dimensions do not inherently scale with m for any configuration, while the MED-C $O(k^2 \log m)$ demonstrates logarithmic dependency empirically in a constrained but achievable setup. We agree that the centroid assumption is special and not universal (e.g., it assumes queries as centroids, limiting applicability to complex queries), but it suffices to contradict prior claims of polynomial limits in geometric space, and *is never intended to be applied to real-world scenarios*. For practical extensions, we discuss hierarchical data (e.g., clustering) as a potential bridge, reducing the effective dimensionality in reranking scenarios from $O(k^2 \log m)$ to $O(k \log m)$ in the best case.

- **Tightness and Empirical Validation**: Reviewers (e.g., JMqo, KFqx) inquired about bound tightness and experiment completeness. The MED-C upper bound differs from MED's $\Theta(k)$ by a $k \log m$ factor—a modest logarithmic gap in $m$ but linear in $k$. Our simulations validate the $O(\log m)$ dependency. Discussions about related works suggested that removing the additional $k$ factor might require additional time complexity. While not scalable for large $m$ (as it checks combinations), this aligns with our approximability focus; future work could explore learnable methods. We have added notes on the probabilistic method's "best-case" existential proof and suggestions for revisiting constructions like cyclic polytopes in real-world implementation.

- **Comparison to Prior Work and Motivation**: As highlighted by KFwt and others, we have strengthened motivations by contrasting our explicit $m$- and $k$-dependent bounds with prior sign-rank hardness and empirical polynomial fittings. Our results suggest limitations of embedding-based systems stem from learnability, not geometry, offering optimism for low-dimensional retrieval.

Best regards,

---

### Meta-Review · Area_Chair_qQvu · 2025-12-16

**Summary:**

This paper presents tight bounds for dimensions required for embedding-based retrieval. The novelty and the importance of the result as theoretical underpinnings for the capability of this commonly used retrieval models are broadly appreciated by the reviewers. However, the paper falls short on the following aspects:

- Quality of presentation and consistency. This is a common concern raised by most reviewers. The paper appears to have been *significantly* revised during the rebuttal period to improve clarity, which raises doubts on the readiness of the submission. Moreover, I still find some loose ends in the current version (e.g. Example 3.1, x is in R^{d+1} instead of R^d; Theorem 3.1, not specified what values t_1, ..., t_m should take: maybe arbitrary?).

- Scope theory vs practice. A few reviewers pushed for experiments with real networks. However, the authors argued that this work proves existence, and that "learnability" is outside the paper's scope. While I understand that not all papers need to cover every aspect of the problem, the choice of scope in this case does weaken the contribution. First, the theoretical results are primarily obtained from applying existing mathematical tools, instead of introducing new technical lemmas and proof techniques that can be of separate interest. Furthermore, a learnability study could have complemented well the theory part in understanding whether the tight bounds are achievable in practice. Such a study may be experimental instead of theoretical, which is standard in the literature on embeddings, and may benefit the paper substentially.

- The paper is missing an important prior work of Guo et al. (see [a]), which mostly trivializes the results on MED-C. In particular, Guo et. al. in their Section 2.2 studied exactly the same problem as in this submission. They constructed embeddings in essentially the same way as in the MED-C proofs, where the doc embeddings were chosen as random vectors and query embeddings as the mean of matching doc embeddings. The results in Guo et al. also appears stronger, requiring only d =  k*log(k*m) instead of k^2 log(m) as in this paper. A similar analysis is also presented in a recent work [b].

Overall, I find the paper to present an exciting result that drastically reduces the previously known bound (in [a]) of k*log(k*m) to a new bound 2*k. The result is significant in making embedding models theoretically grounded for retrieval tasks, which can have profound implications to the information retrieval, retrieval augmented generation, generative retrieval communities.

However, the paper is missing critical references that invalidate an important part of the submission. It could also benefit from some empirical studies on the learnability of such embeddings -- which can shed lights on the gap between the theoretically known tight bound with what's actually achievable in practical learning. Finally, the paper appears to have been prepared in a rush with may typos and formatting issues that may have hindered a thorough evaluation from the reviewers.

Once the issues are fixed, I am hopeful that this could be a highly influential work.

[a] C. Guo, A. Mousavi, X. Wu, D. N. Holtmann-Rice, S. Kale, S. Reddi, and S. Kumar. Breaking the glass ceiling for embedding-based classifiers for large output spaces. NeurIPS, 2019.
[b] Hierarchical Retrieval: The Geometry and a Pretrain-Finetune Recipe, NeurIPS 2025.

**Reviewer Concerns:**

# KFwt:

- Typos: May still have lingering typos

- More discussion about the earlier work: Added and addressed.

- Real experiments: Author argues to be not in scope. Unclear if the reviewer would have been satisfied.

# JMqo:

- Tightness of the MAED bounds: tigher bounds appear to exist in [a, b]

- Other clarification questions that the authors have addressed.

# txVp

Reviewer questioned the "achievable" claim and practical implications in general. The author argued that it is out of the scope.

# KFqx

Manuscript is quite unpolished: This is their major concern. The authors have significantly revised their paper, and the reviewer likely needs to read and reevaluate the paper, which unforunately is impossible.

# Dok3

- The exposition of the paper is quite cryptic sometimes: This has been improved but again reflect the fact that the paper was not entirely ready at the time of submission, and it is unclear that with the substential revision whether the reviewer would have changed their mind.

- Practic value: This was again addressed by not in scope argument, and unclear if the reviewer would have been satisfied with that.

**Reviewer Scores:**

KFwt: Reviewer had been leaning positive. They likely keep their score of 6.

JMqo: Reviewer was already quite positive, likely keeping their score of 8.

txVp: Reveiwer was positive in rating, but had a lot of comments on the algorithmic aspect that were argued as out of scope. Unclear how much weight the reviewer would have put on those aspects in their final rating.

KFqx: Reviewer was quite negative, likely due to the fact that the paper was not very polished enough for an evaluation. Now with the revision, I am not sure if reviewer would had been able to reevaluate the paper during rebuttal. They may had kept the score to 2 or increased it to 4.

Dok3: Similar to KFqx, they may have raised to 5 if they reevaluate the submission.

---

### Decision · Program_Chairs · 2026-01-26

Reject